# Temporary Agency Work and Well-Being—The Mediating Role of Job Insecurity

**DOI:** 10.3390/ijerph182111154

**Published:** 2021-10-23

**Authors:** Birgit Thomson, Lena Hünefeld

**Affiliations:** Bundesanstalt für Arbeitsschutz und Arbeitsmedizin (BAuA/Federal Institute for Occupational Safety and Health), Friedrich-Henkel-Weg 1-25, D-44149 Dortmund, Germany; huenefeld.lena@baua.bund.de

**Keywords:** temporary agency employment, job insecurity, job and life satisfaction, psychosomatic complaints, well-being

## Abstract

Organisations use non-standard employment as a means of flexibility and reduction of fixed costs. An increasingly growing group of employees are self-employed, have work contracts such as part-time and temporary contracts or are employed by a temporary agency, a development catalysed by the COVID pandemic. Whereas there is some evidence that temporary work might affect health via job insecurity (JI) there are hardly any studies focussing on the effects and mechanisms of temporary agency work (TAW). This study sheds light on TAW’s potential health impact and the role of JI in this respect using a mediation analysis. Based on the BIBB/BAuA-Employment Survey 2018 (N = 20.021, representative of the German working population), we analysed the direct effect of TAW on cognitive and psychosomatic aspects of well-being. In particular, we considered JI as mediator for this association. In line with the potentially detrimental effects of temporary employment on well-being, we found that TAW was related to unfavourable outcomes in terms of job satisfaction, general health status and musculoskeletal complaints. JI partially mediated all three underlying associations. Organisations need to be flexible and adaptable. However, by using temporary agency employment as a means to achieve this flexibility, managers and leaders should be aware that it is related to unfavourable well-being and hence hidden costs. In using this type of employment, both the temporary work agency and the user company should consider these health risks by providing health care, options for increasing the temporary agency workers (TA), workers employability, and equal treatment between permanent and TA workers at the actual workplace.

## 1. Background, Objective and Contribution

Non-standard employment (NSE) is a fast growing global phenomenon that affects more than one-third of the worldwide workforce [1]. Temporary agency work (TAW) is one example of NSE. It is assumed to be associated with precarious labour and life conditions [2,3]. Among the various forms of flexible employment, TAW is the most dynamically increasing type with currently about 54 million temporary agency workers/TA workers [4]. Even though TAW has also grown continuously in the EU over the recent years [5,6], it developed differently in various national economies. For example, in Ireland, the proportion of TA workers almost tripled between 2011 and 2020 (2011: 0.9%, 2020: 2.5%), while in Germany it doubled over the same period (2011: 2.4%; 2020: 5.4%). In 2020, 2.6% of the overall European Union (EU 27) workforce was employed in TAW [6], which on first glance seems to be a low proportion. NSE accounts for about 40% of total employment in the European OECD countries in the sectors most affected by containment measures (e.g., short-time work, loss of earnings) [7]. The COVID pandemic has increased inequalities by affecting vulnerable groups disproportionately [8]. Based on a recent comprehensive literature review Kniffin et al. (2020) [9] summarized presumable effects of the pandemic and pointed out that unemployment and JI will become more predominant as a result. Particularly in sectors such as gastronomy, tourism, service and non-food retail, the number of TA workers who were given notice has already increased because of the pandemic [10]. Moreover, in previous similar shocks such as the financial crisis in 2008, precariously employed persons such as TA workers tended to be even worse off due to the crisis of increasing social inequality [11].

In this respect, it seems unlikely that the formal employment contract in itself determines individual outcomes such as job attitudes or well-being. It is rather more realistic that underlying mechanisms related to working conditions should be analyzed in order to identify options to mitigate potential detrimental outcomes.

One of the most important work stressors, particularly in times of dynamic change, is job insecurity (JI), defined as peoples’ fear of losing their job and their source of income [12,13]. Given that the employment contract type is known to be one of the most critical factors affecting JI [14,15] it is suggestive that JI might mediate or account for the association between TAW and its potential well-being outcomes. So far, however, there is very little research support this assumption. Even though there is evidence that JI mediates the association between other forms of NSE (e.g., temporary employment) and negative individual impacts such as well-being and health [16,17], there are hardly any similar studies available for TAW [18,19,20]. In summary, TAW is a fast growing form of flexible employment globally. The dynamic of this growth has been exacerbated by the COVID 19 pandemic, which increased labour market segmentation and social inequalities concerning a large number of employees. It increased JI as one of the most important work stressors implying physical and particularly mental health risks for workers experiencing high uncertainty. There is, however, a research gap in terms of the association between TAW and both well-being and health, primarily due to the lack of eligible or sufficiently sized TA worker samples. Flexible workforces and TA workers in particular, therefore, become an increasingly important group, whose working conditions and the potential impact of these conditions are insufficiently covered by research. Against this background, the objective of our contribution is to tackle the research questions on TA workers’ well-being as opposed to workers in standard employment and to investigate JI’s potential role as a mediator of this association.

We contribute to the existing literature by tackling this important gap in TAW research. Even though TAW is a growing phenomenon, the actual number of TA workers in the various national economies is still small. Differently from most studies which draw on small and often convenient samples, we contribute to the research of TA workers health and well-being based on a current, large national survey representative of the German working population [21]. The overall sample size (see sample description in the method part) results in a (relatively to other studies) remarkable sub-sample of about 400 TA workers, which allows for valid results in terms of our research question. By presenting empirical evidence that TAW has a negative impact on well-being and that JI mediates the TAW-well-being relationship, we contribute to the discussion of which workplace measures might be important prevention measures with regard to TA workers health.

## 2. Theory and Hypotheses

NSE might comprise varied forms of employment among countries with different labour markets and laws. Nevertheless, the term “NSE” usually applies to cover those work arrangements that fall outside the realm of the standard employment relationship. TAW as one NSE example is based on a tripartite employment relationship involving a worker, a company acting as a TA and a user company. Compared to standard or temporary employment the triangular nature of TAW results in two different employer-relationships for the TA worker. He or she relies on the TA in terms of selection, training, pay or contract duration, whereas with respect to the actual work demands/conditions, resources, social networks etc., the TA worker is confronted with the user company’s working conditions [22].

The concept primarily meets the user company’s interests as regards flexible labour in order to quickly adapt to external demands or, more specifically, to have specialized workers available in the short-term, while cutting fixed costs. Likewise, employees might find this form of employment suitable in terms of flexible contractual arrangements or work-life-balance issues [23,24]. In her systematic review on NSE’s health impact, Hünefeld (2016) [25] found that whether or not individuals deliberately chose non-standard contract forms has an influence on the employment’s potential health impact. If the contract form is non-preferential, this will reinforce potential negative health and well-being effects. In the majority of EU national economies, TAW is not workers’ preferred employment form but rather the result of a lack of alternatives [26]. Hence, concerns in terms of potentially negative impacts of this employment form for individuals in terms of attitudes, well-being and health arise. Although TA workers are a minority in the labour market, there are important reasons why research should focus on this group. TAW contracts are typically based on short utilisation times by the user companies in order to fulfil new work requirements, so that TA workers frequently have to change workplaces [27]. TAW is assumed to be the most insecure contractual type in Europe [28], as due to the short-term character of the various employment aspects, even TA workers who have permanent contracts with the TA perceive more quantitative and qualitative JI than the user organisation’s employees [29].

In terms of the TAW-well-being association segmentation theories, i.e., concepts explaining the mechanisms of labour markets and their dilimitable parts, we refer to the high uncertainty levels involved in TAW employment with its detrimental outcomes [30,31]. The latent deprivation model [32] claims that work, as the source for economic and social needs, is one of the most crucial aspects in peoples lives. Work provides resources such as income, social status, social networks and structure, which are the basis for individual development and personal thriving [33].The lack of these resources due to uncertainty in terms of having a job and income is supposed to be highly detrimental for individuals [32]. Conservation of resources theory/COR [34,35] predicts that people experience stress, if such key resources are at stake, if the person loses these resources or if he/she cannot regain the resources despite major efforts [35]. COR refers to various resource categories such as objects (e.g., housing, food), personal traits (e.g., resilience, self-efficacy, optimism), energy resources (e.g., money, social support network), context resources (e.g., cultural context) and conditions such as stable and secure employment or status [35]. Clearly, the characteristics of TAW as described above are a threat to a whole range of these resources and accordingly imply stress and risks for well-being and health.

In predicting TAW’s detrimental well-being impact segmentation theories, we also refer to TA workers comparably unfavourable working conditions [18,36]. Compared to the employer’s own workforce, TA workers often receive lower pay and fewer benefits, they can only infrequently participate in career planning and training, atypically hold lower professional ranks [36], get less occupational health and safety trainings, have less access to health promotion measures [37], and work under stressful and hazardous conditions [29,38,39,40]. Employees exposed to these occupational hazards and risk factors are more likely to enter loss spirals in terms of resources as COR predicts persons with fewer resources are more vulnerable to resource loss, which can be significant, happen fast and imply consequences beyond the immediate perception of stress [33].

Studies have not yet established a conclusive and consistent link between the types of contract in terms of NSE and unfavourable individual outcomes by explaining the underlying mechanism [16,41]. Apart from the theoretical considerations of COR and segmentation theory we can, however, derive hypotheses as regards the health impact of TAW based on two current literature reviews. Analysing 106 studies, Hünefeld (2016) [25] concluded that TAW is detrimental in terms of job satisfaction, well-being and health—particularly mental health. Different from the inconsistent results in terms of the negative impact of other forms of atypical work (temporary work, part time work, self-employment, multiple jobs), TAW was found to be consistently detrimental for well-being and health. The review reported medium sized effects [42] between TAW and depression as well as small to medium sized effects for general health, burnout, job satisfaction, motivation and physical health. In this study, we analyse lack of job satisfaction (amongst other factors) as detrimental to well-being. The concept refers to a person’s perception whether a work context meets his or her demands and needs or not [43]. Negative perceptions in this respect are closely related to health impairment (Burnout, Depression, anxiety, mental health [44]). Based on 24 studies, Hünefeld et al. (2020) [20] found that TAW is unfavourably related to job satisfaction. Due to theoretical considerations and these current and consistent findings, we postulate the following.

**Hypothesis** **1** **(H1).**
*TAW will be positively related to well-being impairment.*


### 2.1. JI and Well-Being

One of this study’s main objectives is to analyse JI’s role in terms of detrimental TAW well-being outcomes. JI refers to the uncertainty as regards workers’ jobs future existence (quantitative JI) or important job characteristics (qualitative JI [12,45]). Both of these aspects have proven to be detrimental in terms of individual outcomes (for quantitative JI see [12,15,46] for qualitative JI see [47,48,49]). In this study, we focus on the quantitative aspect of JI. It captures employee uncertainty about whether they will keep their job in the future, and is defined as “the perception of potential threat or continuity in his or her current job” [50]. JI definitions include two common subjective perceptions: (1) that one might face unemployment, and (2) that JI is undesirable and stressful [12].

In order to explain the detrimental health and well-being impact, JI literature generally draws on either stress-theories [12,46] or social exchange theories, and psychological contract theory/PCT in particular [51,52]. If individuals are at risk of losing their jobs, they can feel increased fear and stress that keeps them from applying effective coping strategies to resolve the situation [53,54]. Like all stressors, JI is related to strain and negative health effects [54,55]. In terms of PCT [51], employees and employers have mutual expectations, of which the provision of a secure job is among the most important employee [56]. Without it, the contract will—from the individual’s perspective—be breached or violated, which can then have an impact on emotions and well-being [57]. In the German economic context, TAW is generally characterised by short contract durations between the employee and the TA [58]. For that reason, job security might actually not be one of the psychological contract’s contents for TA workers. Studies have found that JI is problematic for standard employees but not for temporary contracts in terms of job satisfaction and organisational commitment [38,59].

On the other hand, TAW is also assumed to be a potential stepping-stone into permanent employment with the user company. Thus, despite the absence of a contractual relationship between the employee and the user company, TA workers still might hope for or expect secure employment. Therefore, the PCT might be a useful alternative theory background for our research question.

In terms of the JI-well-being relationship, there is broad empirical evidence that JI is a profound work stressor and implies breach and violation of the psychological contract. Sverke et al. (2002) [12] conducted a meta-analysis on the potentially negative impact of JI on work attitudes, behaviour and health. As for health they found that JI has unfavourable effects on both physical and mental health, with more consistent and stronger effects on mental health, as confirmed by Cheng & Chan (2008) [46], who replicated this meta-analysis based on a more recent and larger set of data. More recently, Köper & Gerstenberg (2016) [15] confirmed, in a comprehensive systematic review, JI’s negative impact on subjectively estimated general health, physical health (cardio-vascular diseases, muscular skeletal diseases, metabolism related diseases, infections) and particularly mental health (psychological well-being, psychological and behavioural diseases, burnout, general mental health). Associations between JI and mental health were both more consistent and stronger compared to physical health. Against the background of this broad empirical evidence, we hypothesise:

**Hypothesis** **2** **(H2).**
*JI will be positively related to well-being impairment.*


### 2.2. TAW and JI

JI has been repeatedly described as an inherent feature of temporary employment arrangements [38] resulting in the notion that the contract type might be one of the most crucial factors affecting JI. Even though TA workers often have a work contract which is formally not time-restricted, they are made redundant if the TA cannot provide further employment [20]. Hence, just like workers in temporary employment, TA workers might experience low job control and low predictability [60]. TAW is a highly dynamic employment sector implying an increased risk for unemployment, if there is little demand for TA workers by the user companies [58]. Accordingly, results from an extensive review of TAW and occupational health/safety indicate JI as a major psychosocial issue in the group of TA workers [29]. Keim, Landis, Pierce and Earnest (2014) [14] identify aspects which predict JI. The type of employment contract, proved to be significantly related to JI and to dissatisfaction with job security [24]. Debus et al. (2014) [61] also showed that NSE predicted JI and interacted significantly with personal predictors (locus of control), implying that the type of contract affects the perception of JI both directly and indirectly by reducing workers’ control perceptions. In terms of differences between workers in NSE and standard employment, Schalk, de Jong, Rigotti, Mohr Peiró and Cabaler (2010) [62] found that employers are more likely to provide important resources such as job security to their permanent employees. Given these empirical findings, we hypothesize:

**Hypothesis** **3** **(H3).**
*TAW will be related to JI such that individuals in temporary agency employment express more JI than permanently employed individuals.*


### 2.3. The Role of Job Insecurity in the Association of Temporary Agency Work and Well-Being

In recent years, studies have specifically reported problematic working conditions in NSE [39,63,64,65]. Rises in general uncertainty and JI have been an important feature in this respect. Both TAW and JI are hypothesized to be detrimental for well-being and health (see H1 and H2). Against this background, we analyse JI’s role in the TAW-well-being association. De Cuyper & De Witte (2007) [60] investigated possible interaction effects between contract type (temporary employment) and JI for well-being outcomes. They found that the interaction between contract type and JI added significant variance in job satisfaction and organisational commitment.

Keim et al. (2014) [14] found that the formal contract type was a predictor for JI, which in itself is one of the most important stressors in the workplace [45], suggesting a mediating role of JI for the health impact of TAW. In their meta-analysis about TAW and well-being, Hünefeld et al. (2020) [20] also found hints for the mediating role of JI for the TAW- well-being association. Either the association of TAW and job satisfaction did not persist after adjustment for JI or a significant reduction in explained well-being variance occurred after controlling for JI. Therefore, we postulate:

**Hypothesis** **4** **(H4).**
*JI will mediate the association of TAW and well-being.*


## 3. Methods

We based our analyses on the BIBB/BAuA-Employment Survey 2018, which is representative of the German working population [21]. 20,012 employees older than 15 years and working at least 10 h per week completed a highly standardised interview conducted by trained interviewers by phone [21]. Since the question about TAW was only addressed to dependent employees (as opposed to self-employed ones), our basic sample resulted in *n* = 16,793. Only participants who provided information on JI and all relevant covariates (e.g., gender, age, educational level) were included, reducing the final sample to *n* = 15,658 (weighted; 46.2% female) with a mean age of 43.4 (SD = 11.9) and 398 (2.5%) TA workers (see Table 1). *TAW* was measured by the single-item “Are you employed by a temporary employment agency that places you with other companies?” TAW was coded as 1, all other employment forms as 0. We measured JI by the item ‘How high do you estimate the risk that the contract will not be extended or released in the near future?’ with a four-point Likert scale (1 being that there is no danger and 4 being very high).

We captured different facets of well-being to cover both short- and long-term consequences of TAW [44]: Job (dis)satisfaction as regards the main occupational activity was measured by a single-item measure of global job (dis)satisfaction on a four-point Likert scale from 1 (very satisfied) to 4 (unsatisfied). Single-item measures of global job (dis)satisfaction have been shown to be as reliable as multiple-item measures of (dis)satisfaction [66]. Whether or not an individual is satisfied with his or her job is based on a comparison of actual and favoured working conditions and can be regarded as a salient aspect of psychological well-being [67]. The construct is also often categorized as a well-being or health outcome [23,31].

The self-rated general health status was gathered based on the single question “How would you describe your general status of health?” and assessed based on a five-point Likert scale from 1 (excellent) to 5 (bad). Using a single-item-measure for self-rated general health is a common procedure. As another health measure, we captured eight items for musculoskeletal complaints (e.g., pain in the back or neck). The measure covers various health dimensions, with focus on functioning and diseases [68]. The persons were asked whether they experienced the eight complaints. We then counted the answers and created a sum index ranging from 0 (no complaints) to 8 (all complaints). This sum index was successfully used in a previous study [69].

We controlled for gender (binary), age (metrical), educational level (ISCED 1997/UNESCO United Nations Educational, Scientific and Cultural Organization, 2003 [70] depicted by the categories low/middle vs. high), working time (part time vs. full time) and industrial sector (industry vs. public sector/craft/service sector/other sectors; for further Information about the control variables see Table A1). There is evidence that these variables are relevant confounders in terms of their influence on JI and/or well-being As for the gender aspect, studies indicate that women are both more likely to report/experience unfavourable health conditions [71,72] and job insecurity [73,74]. There are also several studies tackling the importance of gender for the association of TAW and well-being [19,24].

In terms of age, studies point to positive associations between increasing age and job insecurity [75,76] as well as health.

The educational level is another relevant confounder given the evidence of the association between low educational levels and both TAW, JI and poor health [77,78]. Finally, the industrial sector is an eligible variable as proxy for various working conditions and organizational settings that can influence JI [15] and health [79].

We conducted regression- and mediation analyses to gather evidence for the hypothesised role of JI for the TAW-well-being- association (see mediation path model in Figure 1). In order to estimate and test the coefficients of this path model, OLS regression was operated in two models (model 1: univariate, model 2: adjusted for gender, age, educational level, working time and industrial sector). For testing the mediation hypothesis, we applied the PROCESS Macro model 4 and bootstrapping with a 95% confidence interval (CI) (2013). In terms of the mediation the analysis differentiates the direct (γ0), indirect (αβ) and total effect (γ = γ0 + αβ) of TAW (X) on well-being (Y). The indirect effect was considered significant if its 95% bootstrap CIs from 10,000 bootstrap samples did not include zero. PROCESS uses ordinary least squares (OLS) regression to estimate the mediation model. In order to adjust for the violation of homoscedasticity, heteroscedasticity-consistent standard errors were applied (HC3; Davidson-MacKinnon). As weighted data can produce biased standard errors and inefficient estimates in ordinary least squares regressions [80], the unweighted sample of employees (*n* = 15,315) was applied.

## 4. Results

Table 1 depicts means and standard deviations for our variables with significant correlations between both TAW and JI and the well-being variables. Overall, the effect sizes are bigger for the JI-well-being-association than for the correlation of TAW and well-being.

The presentation of results in this section follows the mediation path model in Figure 1 rather than the hypotheses sequence. In the OLS regression of JI on TAW (Table 2), the coefficient α of the underlying path model (see Figure 1) is estimated. Model 1 (univariate) shows positive and statistically significant regression weights. Even after adjustment in model 2, the effects remain almost identical and significant on a 1%-level. TAWs therefore experience JI more frequently than standard employees, even after controlling for gender, age, educational level, working time and industrial sector. The results support H3.

Table 3 shows the results of the OLS regression of TAW and JI for the different indicators of well-being. For the coefficient β of the path model, there are significant effects in the unadjusted models (model 1), and JI was positively related to well-being impairment, supporting H2. These effects remain on the 1%-level of significance even after adjustment in model 2.

Workers exhibit higher job dissatisfaction, poorer general health, and more musculoskeletal complaints as job insecurity increases, even after adjusting for the control variables. For the direct effect through the coefficient γ’, positive and significant effects are shown in model 1 for work satisfaction and musculoskeletal complaints. However, we found no significant effect for general health status. After adjustment in model 2, the regression weights remain constant or increase, and the effect between JI and general health status becomes significant. TA workers show higher dissatisfaction with work, poorer general health status, and more musculoskeletal complaints compared to standard employees (support for H1). The increasing regression weights in model 2 regarding general health status and musculoskeletal complaints indicate that the included covariates are possible moderators for the association of TAW, JI and well-being. Further analyses, in which the covariates were added to the model individually, indicate that age could be a moderator for the association of TAW and general health status and musculoskeletal complaints with the relationship being more detrimental for older employees.

The estimated total effects of TAW on work satisfaction, general health status and musculoskeletal complaints result in positive regression weights for the coefficient γ in the first and second model (Table 3). In both models, the effects are stronger for job satisfaction and complaints than for general health status. For the indirect effect (αβ) of TAW on well-being mediated by job insecurity, positive estimates are shown in the univariate and adjusted models (Table 3). The bootstrapping process indicates a mediation effect by JI, as the 95% bootstrap CIs from 10,000 bootstrap samples did not include zero. Accordingly, the relationship between TAW and well-being is partially mediated by JI, supporting H4. In summary, our results support all four hypotheses and respectively the assumed mediation model, so that JI is part of the mechanism in the association of TAW and well-being impairment.

## 5. Discussion

Recently, the COVID 19 Pandemic has drastically accelerated work related phenomena such as inequality, unemployment, and has increased precarious work and job insecurity [9]. As described in the introduction and theory section, this implies resource depletion [34,35], and particularly affects those persons who dispose of fewer resources and are hence more vulnerable to further resource loss. As we have argued, under- or insecure employment, income inadequacy or job insecurity are related to TAW and place TA workers in a deprived situation.

This highlights the relevance of analysing both the potentially detrimental individual outcomes of employment forms, which might boost these problems and the mechanisms underlying the TAW-outcome-relation in order to derive preventive measures in both the TA and the user company. In this contribution, we shed light on question of TAW’s well-being impact and the mediating role of JI. We could show that TAW and JI were both unfavourably related to well-being, that TA workers experience more JI compared to workers in standard employment, and that JI mediates the relationship between TAW and well-being. This issue has hardly been tackled in the literature [16,17,18,19], mostly due to non-existent samples or insufficient sample sizes of TA workers. In their systematic review on TAW and well-being, Hünefeld et al. [20] point out that even though there is evidence for JI’s role in the association of temporary employment and well-being, the equivalent evidence for TAW as more precarious form of work is missing. Based on a representative employee survey which provides a large sub-sample of approximately 400 TA workers, we attempted to close this research gap. We found that just like temporary employment, TAW relates to unfavourable well-being and psychosomatic health outcomes partially mediated by JI, i.e., the fear of losing one’s job and income source. Despite small effect sizes, JI explained additional well-being variance.

On first glance, it might be surprising that for TA workers JI is such an important detrimental feature and that they experience higher JI than workers with standard contracts. After all, the TA workers’ contracts with their employer i.e., the TAW is not by definition timely restricted, so that TA workers should experience less JI than those in other precarious employment, such as, for instance, temporary workers. Obviously, the answer for the dominant role of JI lies in the precarious and insecure character of TAW [2,28], resulting in a great deal of uncertainty as regards continuity of employment, and of work characteristics and income, which can start or reinforce resource depletion and loss spirals [35].

If there is little demand for TA workers by the user companies, TAs tend to give notice to the TA workers short-term. TAW contracts are often based on short utilisation times within the user company so that TA workers are continuously confronted with different workplaces and new demands [27]. Accordingly, it is particularly important for TA workers to keep their skills up to date as this is a basic condition for being able to take on new tasks. However, Håkansson & Isidorsson (2016) [29] point out that TA workers in particular are confronted with a lack of training opportunities, which increases their feeling of JI additionally.

JI due to disruptive employment is one of the most important work stressors [45,15). De Witte (2010) [45] presumed that JI might even be as detrimental for well-being and health as actual unemployment. In line with the importance of JI as workplace stressor, our study showed that it was significantly related to unfavourable well-being after controlling for relevant demographic characteristics such as education, gender, age and working hours per week. But even beyond the aspects of uncertainty and unfavourable working conditions, TAW is widely assumed to be more unfavourable than permanent work arrangements, as it is often associated with precarious life situations [2,39,81,82]. The reasons for this lie in the unequal and unfair treatment of TA workers in comparison to core workers [27]. Even if TAW falls under the same national frame of labour law legislation, TA workers are generally treated differently in the user company in terms of payment, training opportunities, health promotion offers and working conditions [20]. Hünefeld & Gerstenberg (2018) [65] compared a number of concrete working conditions of TA workers and those in standard employment. They found that TA workers have fewer important resources such as work latitude, planning options in terms of working time and breaks, participation options, social support, and voice. At the same time they face more unfavourable working conditions like untypical working time and more difficult physical working environments, unsuitable workloads (too high or too low), and monotony. The relevance of other work stressors in TAW explains the fact that JI only partially mediated the association of TAW and well-being, implying that future research should also consider further potential mediators.

Another reason why JI as a feature of TAW was detrimental in this sample might be the German cultural background. There are not many studies available which focus on cultural differences in the perception and appraisal of JI. As one of the few studies available on this issue, Otto et al. (2015) [48] differentiate JI’s impact in “masculine” and “feminine” cultures. In Germany with its “masculine” culture, work has a particularly high value for peoples’ identity and self-esteem. The authors found that the negative impact of JI on work attitudes and health was stronger for German employees than for employees in Scandinavian countries with their “Nordic Model” and a stronger focus on cooperation as well as care and support. Hence, for German nationals JI does not only threaten the employment as such but probably also their social status and their self-concept.

As TA workers face high levels of uncertainty combined with unfavourable working and life conditions, both the TA and the user company have responsibilities in tackling these aspects and should try to mitigate the job related uncertainty by considering TA workers’ qualification and employability

## 6. Limitations and Future Research Recommendations

We are aware that our methodical approach has some limitations. First, we based our reasoning on cross sectional data, which despite their broad use are more and more disdained, as it does not allow for deriving causal relationships. In this regard, applying mediation analysis is also problematic as it implies causal directions-in this case, from TAW to well-being-impairment via JI. However, tackling this issue, Spector (2019) [83] recently published a methodological paper on the use of cross sectional data. He argues that the ability of longitudinal data as a basis on which causal conclusion can be derived has been overestimated and that in most cases longitudinal data actually bear only slight advantages over cross sectional data in many research designs, as the latter are as eligible to exclude explanations for associations alternative to the hypothesis. In order to improve cross sectional designs, he suggests amongst other things to control for other relevant influences, which we did by controlling for relevant aspects in terms of well-being and health, i.e., gender, age, educational level, working time and the industrial sector. In line with Spector’s (2019) [83] reasoning, we do not suggest causal relationships between TAW and well-being. Rather, our results show that TAW and health impairment are related and that JI explains additional variance in this association. Based on other studies it is, however, suggestive to assume that TAW is an antecedent of JI [14] rather than vice versa. After all, the reversed causality assumption that TA workers chose this employment form because they feel uncertain in terms of their future employment (JI) is not sensible. Moreover, there are few studies available which provide hints for the causal long-term health effect of JI. Hellgren & Sverke (2003) [84] showed that the cross-lagged effect of JI on mental health was still significant after one year, whereas the reversed causality association was not significant. As the authors could not find the same results for physical health, the question of JI-health causality remains to be tackled in future studies.

A second methodological drawback lies in the nature of our measures. In this respect, the advantage of a big TAW subsample comes at the cost of the representative survey’s characteristics with its broad approach. Like most other national surveys, it is largely based on single items rather than validated, multi-faceted constructs. Yet, as far as TAW (the central aspect of this study) is concerned, there is (to our knowledge) no validated standard-measure available anyway. For future research it is, therefore, essential to find appropriate concepts to gather TAW’s various aspects such as contract form, contract duration, or occupational status. TAW in particular needs to be measured in a separated manner from other forms of temporary work. [23,82]. As far as JI is concerned, Sverke et al. (2002) [12] analysed whether or not the type of measure moderates the association of JI and negative outcomes. They found that based on single measure items for JI its negative impact is likely underestimated, which is related to another limitation, which is the small effect sizes reported in our results. Based on the results by Sverke et al., (2002) [12] the small effect sizes in our sample might in parts result from the single item measures.

JI as mediator explained additional variance, even though we controlled for the most relevant confounders (see method section). This procedure is adequate (Spector, 2019) [83], and constitutes an advantage compared to many other NSE studies. Yet it will have reduced variance for the other model variables. By and large, however, the effect sizes in our study are in line with results from recent systematic literature reviews, which likewise report small effect sizes of both TAW and JI on health [25,65]. Based on more than 220 studies on the relationship of JI and health, Köper & Gerstenberg, (2016) [15] found average correlations (based on weighted samples) between JI and health of ${\bar{r}}_{ij}=0.20$ for mental and ${\bar{r}}_{ij}=0.14$ for physical health. They argued that these effects, despite being small, are still highly relevant in terms of attributive risks [85]. At an assumed JI average prevalence rate of 16% in the EU, effect sizes for mental health would imply that 9% of mental health impairment relates to JI.

A last aspect in term of limitations is the focus on just one national economy with its specific laws and regulations for TAW. As described above, the German “masculine” culture [48] would suggest that the individual impact of TAW and JI are more severe compared to other countries. Future studies should, therefore, include international comparisons tackling these national differences and their meaning for TAW’s individual impact.

## 7. Conclusions for Practical and Policy Implications

Organisations doubtlessly need to be flexible and adaptable, particularly in times of ongoing global phenomena such as digitalisation, global division of labour etc. Yet on the societal level, TAW is considered as precarious work and is associated with income insecurity, lack of coverage by national social networks, employment instability, low social status and reduced employability [86,87]. The COVID pandemic has clearly reinforced economic stressors and potential resource depletion for TA workers [9,33]. Considering this notion, research after the pandemic should soundly analyse the “new normal” of working conditions. The pandemic will have been a catalyst for further digitalization and more flexible work characteristics in terms of remote or hybrid work or timely flexible work, often at the cost of more work intensification. In industry sectors where despite the pandemic the reduction of interpersonal contact was not an option, implying more safety risks (in sectors such as hospitality, tourism, service, and retail [10]), other safety and psychosocial risks emerged [9] and have to been dealt with in a post pandemic scenario. These diverse experiences in organizations, sectors and national economies will continue, affect organisational structures and processes and reinforce job related inequalities [11]. For sectors and worker groups with increased flexibility demands, their related psychosocial risks and the combination of psychosocial and safety risks, options to prevent negative individual impacts of “new normal” working conditions will have to differ. Considering well-being and health risks, particularly of those employees with flexible contracts such as TAW, these should become a stronger point on both the Tas’ and user companies’ agenda.

On an individual level, TAW is related to negative a well-being impact via JI, as we have shown in this study. In order to improve working conditions for TA workers both the TA and the user company should take measures to mitigate the potential negative effects of uncertainty and poor working conditions. In this respect, Menatta et al. (2021) [22] recently confirmed that well-being/job satisfaction amongst TA workers was higher if they felt affectively committed to both the TA and the user company. However, if employees were unilaterally committed to only one of the two involved organisations, the commitment with the user company seemed to be of greater importance to well-being. The authors’ explanation for this referred to the fact that in terms of day to day work experiences (working conditions, social network), the user company’s impact is more meaningful, whereas the TA-TA worker relations are, rather, reduced to administrative issues. The authors conclude that it is therefore particularly important for the TA to improve TA workers’ skills and employability in order to increase affective commitment to the TA, which then in turn increases commitment to the user company. The authors also recommended considering the implementation of an “on site” manager if the TA employs a high number of TA workers in the same user company. This person could make sure that the TA can monitor the employees and be present for on-site problems.

In terms of structural working conditions within the user company, TA workers should be treated just like the rest of the staff in terms of equal pay and treatment for equal work [82] and socially integration [88]. Leadership behaviour is crucial for peoples’ well-being at work in general and particularly for non-standard employees [89]. In their review about the health impact of JI, Köper & Gerstenberg (2016) [15] summarised the practical recommendations of the underlying studies as to how to mitigate negative JI impact for individuals, which is relevant in the TAW context as JI explains the mechanism between TAW and well-being. The authors categorised measures on the strategic, the operational and individual level of an organisation. On the strategic level they recommended to rethink whether JI -increasing measures such as restructuring or NSE are actually necessary, and if so, to tackle specific stressors such as JI or work intensification. For the operational level, the underlying studies referred to communication and transparency, particularly in terms of contents of the psychological contract, organisational justice and increased awareness as for employees who are particularly at risk of detrimental individual outcomes. On the individual level, the included studies tackled the improvement of employability options in order to prepare employees at risk of losing their job for the external job market and for the provision of stress prevention measures. As a concluding remark, Köper & Gerstenberg (2016) [15] point out that flexibility and cost reduction options for organisations can come at the cost of individuals’ health. Individuals end up “between Scylla and Charibdis” as Sinclair et al. (2021) [33] caption their recent contribution on individual impact of economic stressors in the world of work, catalysed by the COVID pandemic. This situation illustrates that policy measures are likewise important beyond work design concepts. In this respect Busk et al. (2015) [90] pointed out that policy regulations also have an effect on job insecurity and job satisfaction of TA workers. In their study, they showed that the deregulation of TAW in Germany in the early 2000s led to lower job satisfaction among male TA workers. This can be explained by decreasing wages and higher perceived JI. The study demonstrated that the regulation of TAW should not only focus on the flexibility options available to companies, but also on the needs of TA workers. In Germany, steps are currently being taken in this direction by considering sector-specific regulations that serve to protect TA workers [91]. Furthermore, the role of trade unions in designing healthy TAW should continue to be strengthened in the future. Against this backdrop, future research is also needed that will explore the effects of the TAW regulations on the worker, the TA, and the user company.

## Figures and Tables

**Figure 1 ijerph-18-11154-f001:**
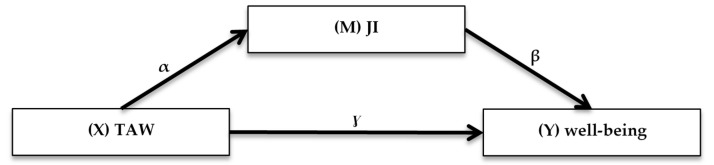
Mediation model.

**Table 1 ijerph-18-11154-t001:** Means (M) and Standard Deviations (SD) and Correlations of the Study Variables.

Variable	M	SD	1	2	3	4	5	6	7	8	9	10
1. TAW ^a^	0.025	0.16										
2. JI	1.53	0.71	0.127 ***									
3. Gender ^b^	0.46	0.50	−0.050 ***	−0.028 ***								
4. Age	43.42	11.94	−0.064 ***	−0.059 ***	0.008							
5. Educational level ^c^	0.33	0.47	−0.054 ***	0.046 ***	−0.030 ***	−0.006						
6. Working time ^d^	0.74	0.44	0.034 ***	−0.013	−0.411 ***	−0.056 ***	0.068 ***					
7. Industrial sector ^e^	0.24	0.43	0.044 ***	0.047 ***	−0.258 ***	0.019 *	0.024 **	0.193 ***				
8. Work (dis)satisfaction	1.81	0.64	0.067 ***	0.211 ***	−0.027 ***	−0.023 **	−0.018 *	0.017 *	−0.01			
9. General health status	2.76	0.87	0.029 ***	0.111 ***	0.032 ***	0.205 ***	−0.129 ***	−0.014	−0.020 *	0.300 ***		
10. Musculoskeletal complaints	2.07	1.99	0.041 ***	0.060 ***	0.111 ***	0.092 ***	−0.198 ***	−0.005	−0.047 ***	0.240 ***	0.458 ***	

Note: Sample size: 15,576 ≤ *n* ≤ 15,658 employees (weighted). ^a^ 0 = no, 1 = yes, ^b^ 0 = male, 1 = female, ^c^ 0 = low/middle, 1 = high, ^d^ 0 = part time, 1 = full time, ^e^ 0 = public sector/craft/service sector/other sectors, 1 = industry. * *p* ≤ 0.05, ** *p* ≤ 0.01, *** *p* < 0.001. (2-tailed).

**Table 2 ijerph-18-11154-t002:** Results of the OLS regression on the association between TAW and JI.

Model 1					Model 2			
	**β**	**95%-CI**	**β^s^**	**R^2^**	**β**	**95%-CI**	**β^s^**	**R^2^**
TAW	0.50 ***	0.42–0.59	0.095	0.09	0.49 ***	0.41–0.58	0.09	0.12

Note: Modell 1: univariate (unweighted), Modell 2 adjusted for gender, age, educational level, working time and industrial sector-Sample size: 15,315 employees (unweighted). β regression coefficient, CI confidence interval, β^s^ standardized regression coefficient, R^2^ coefficient of determination. *** *p* < 0.001.

**Table 3 ijerph-18-11154-t003:** Mediation analyses of TAW, JI, work (dis)satisfaction, general health and musculoskeletal complaints.

Work (dis)satisfaction								
Total Effect Model	Model 1				Model 2			
	β	95%-CI	β^s^	R^2^	β	95%-CI	β^s^	R^2^
TAW	0.29 ***	0.20–0.38	0.46	0.003	0.28 ***	0.19–0.37	0.44	0.005
**Direct Effect Model**								
TAW	0.19 ***	0.10–0.29	0.31	0.04	0.18 ***	0.09–0.28	0.29	0.04
JI	0.18 ***	0.16–0.20	0.20	-	0.18 ***	0.17–0.20	0.20	-
**Indirect Effect Model**								
	β	95%-Boot CI	β^s^	R^2^	β	95%-Boot CI	β^s^	R^2^
JI	0.09	0.07–0.11	0.14	-	0.09	0.07–0.11	0.14	-
**General health status**								
**Total Effect Model**	**Model 1**				**Model 2**			
	β	95%-CI	β^s^	R^2^	β	95%-CI	β^s^	R^2^
TAW	0.15 *	0.04–0.26	0.18	0.0006	0.19 **	0.08–0.30	0.23	0.05
**Direct Effect Model**								
TAW	0.09	−0.02–0.20	0.10	0.01	0.12 *	0.01–0.23	0.14	0.06
JI	0.12 ***	0.10–0.14	0.10	-	0.14 ***	0.12–0.16	0.11	-
**Indirect Effect Mod.**								
	β	95%-Boot CI	β^s^	R^2^	β	95%-Boot CI	β^s^	R^2^
JI	0.06	0.04–0.08	0.07	-	0.07	0.05–0.09	0.08	-
**Musculoskeletal complaints**								
**Total Effect Model**	**Model 1**				**Model 2**			
	β	95%-CI	β^s^	R^2^	β	95%-CI	β^s^	R^2^
TAW	0.47 ***	0.21–0.74	0.25	0.001	0.50 ***	0.25–0.76	0.73	0.01
**Direct Effect Model**								
TAW	0.39 **	0.13–0.65	0.20	0.004	0.39 **	0.14–0.65	0.20	0.07
JI	0.16 ***	0.11–0.21	0.05	-	0.21 ***	0.17–0.25	0.07	-
**Indirect Effect Mod.**								
	β	95%-Boot CI	β^s^	R^2^	β	95%-Boot CI	β^s^	R^2^
	0.08	0.05–0.11	0.04	-	0.10	0.07–0.14	0.05	-

Note: Model 1: univariate, model 2 adjusted for gender, age, educational level, working time and industrial sector-Sample size: 15,256 ≤ *n* ≤ 15,298 employees (unweighted). β regression coefficient, CI confidence interval, Boot-CI Bootstrap confidence interval (number of bootstrap samples for bias corrected bootstrap confidence intervals: 10,000), β^s^ (partially) ^1^ standardized regression coefficient, R^2^ coefficient of determination. * *p* ≤ 0.05, ** *p* ≤ 0.01, *** *p* < 0.001. ^1^ As the regressor (X) is a dichotomous variable, in the indirect effect model as β^s^ partially standardized regression coefficient are reported.

## Data Availability

The data for this study is a representative German employee survey. More information on this survey can be found under the following link https://www.baua.de/DE/Aufgaben/Forschung/Forschungsprojekte/f2417.html (accessed on 21 October 2021).

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
