# Peer review of "Temporary Agency Work and Well-Being—The Mediating Role of Job Insecurity"

_ijerph, 2021, doi:10.3390/ijerph182111154_

Round 1
Reviewer 1 Report
Dear Authors,
I recommend the paper for publication after some revisions, namely:
- Methodology should be shortly mentioned in the Abstract.
- Typically, in the Discussion and Conclusions sections, we use the sources discussed in Introduction / Background , Objective and Contribution. Please explain how your findings relate to the prior literature (reported earlier in the literature review) or develop the Introduction section.
- Referencing style must by adapted to the journal requirements. There are minor lapses here (for example: “Hellgren & Sverke (2003) showed”)
- No consistency in using shortcuts - this needs to be fine-tuned
I do hope you find the comments helpful as you move forward with your conference paper.
Reviewer 2 Report
Overall, this is a really good manuscript examining the relationship between temporary work, job insecurity, and employee well-being. This is a vitally important topic to investigate with constantly evolving updates to the working environment - particularly during the COVID-19 pandemic. The reviewer does have some comments and very minor suggestions for improvement of the manuscript below.
- The introduction is very well organized and the reviewer has no substantial comments for improvement.
- The reviewer would recommend more information on some of the variables used. For instance, how was educational level, working time, and industrial sector measured? It is difficult to obtain the information from the methods and Table 1 in the results. This could help the reader better comprehend the analyses and results that are also found in Tables 2 and 3.
- Overall, the discussion and conclusions are well-organized and supported by the findings from the current study.
Reviewer 3 Report
The reviewed manuscript describes an extremely important problem concerning thousands of employees. The Covid-19 pandemic has only exacerbated their problems, which can also negatively impact physical and mental health. The publication is of great value and recommends its publication in its current form. Congratulations.
Reviewer 4 Report
Thank you for the opportunity to review the manuscript entitled “Temporary Agency Work and Well-Being – the Mediating Role of Job Insecurity” (ijerph-1386277). The authors report a cross sectional study on the association of Temporary Agency Work (TAW) with well-being (i.e., job satisfaction, general health status, and musculoskeletal complaints) and the potential mediation by job insecurity. The authors report TAW to be related to work satisfaction, general health and musculoskeletal complaints, and that these associations were (partially) mediated by job insecurity. Even the manuscript addresses a general interesting topic, it needs fundamental revision. Especially the introduction should be comprehensively revised. The introduction should precisely and clearly present the current state of research on the topic, lead stringently to the research question, and explain why this investigation is important and relevant. I can see that the authors have put a lot of work into the manuscript and I am sorry to say that it needs a major revision.
Some minor points:
In line 28 in the abstract I guess it should be temporary agencywork/ TAW
The abbreviations are confusing and used inconsistent
Line 49 – 50 Temporary agency work (TAW)
Line 51temporary work agency / TA
Line 56 – 57 temporary agency workers / TA
Line 253: what does (ibid) mean?
Line 265: I guess “1 There” should be ”1 there”.
Line 296: What is “(Quelle)”?
Line 378: R2is no measurement of goodness of fit.
In the figure 1, the text is cut.
In table 2, I assume that the standardized regression coefficient of model 1 (0.95) is wrong.
Between line 321 and 322 is an incorrect line break.
I hope my recommendations and comments help to improve the paper.
Round 2
Reviewer 4 Report
I would like to thank the authors for their revision of the manuscript “Temporary Agency Work and Well-Being – the Mediating Role of Job Insecurity” (ijerph-1386277). The quality of the paper improved; however, the introduction might be streamlined more. In addition, a summary of the research gap and research question at the end of the introduction would be helpful.
